# Detecting incidental correlation in multimodal learning via latent variable modeling

**Taro Makino**                                                                                      *taro@nyu.edu*
*NYU Center for Data Science*

**Yixin Wang**                                                                                      *yixinw@umich.edu*
*University of Michigan*

**Krzysztof J. Geras**                                                                              *k.j.geras@nyu.edu*
*NYU Grossman School of Medicine*

**Kyunghyun Cho**                                                                            *kyunghyun.cho@nyu.edu*
*NYU Center for Data Science*
*Prescient Design, Genentech*
*CIFAR LMB*

**Reviewed on OpenReview:** *https://openreview.net/forum?id=QoRo9QmOAr*

## Abstract

Multimodal neural networks often fail to utilize all modalities. They subsequently generalize worse than their unimodal counterparts, or make predictions that only depend on a subset of modalities. We refer to this problem as *modality underutilization*. Existing work has addressed this issue by ensuring that there are no systematic biases in dataset creation, or that our neural network architectures and optimization algorithms are capable of learning modality interactions. We demonstrate that even when these favorable conditions are met, modality underutilization can still occur in the small data regime. To explain this phenomenon, we put forth a concept that we call *incidental correlation*. It is a spurious correlation that emerges in small datasets, despite not being a part of the underlying data generating process (DGP). We develop our argument using a DGP under which multimodal neural networks must utilize all modalities, since all paths between the inputs and target are causal. This represents an idealized scenario that often fails to materialize. Instead, due to incidental correlation, small datasets sampled from this DGP have higher likelihood under an alternative DGP with spurious paths between the inputs and target. Multimodal neural networks that use these spurious paths for prediction fail to utilize all modalities. Given its harmful effects, we propose to detect incidental correlation via latent variable modeling. We specify an identifiable variational autoencoder such that the latent posterior encodes the spurious correlations between the inputs and target. This allows us to interpret the Kullback-Leibler divergence between the latent posterior and prior as the severity of incidental correlation. We use an ablation study to show that identifiability is important in this context, since we derive our conclusions from the latent posterior. Using experiments with synthetic data, as well as with VQA v2.0 and NLVR2, we demonstrate that incidental correlation emerges in the small data regime, and leads to modality underutilization. Practitioners of multimodal learning can use our method to detect whether incidental correlation is present in their datasets, and determine whether they should collect additional data.

# 1 Introduction

Multimodal learning refers to jointly modeling data from different modalities, such as images, speech, and text (Ngiam et al., 2011; Baltrusaitis et al., 2019; Liang et al., 2022). While the goal is to learn modality interactions that are useful for downstream tasks, multimodal neural networks often generalize worse than their unimodal counterparts (Jabri et al., 2016; Poliak et al., 2018; Jean & Cho, 2019; Wang et al., 2020a; Wu et al., 2020; 2022), or make predictions that only depend on a subset of modalities (Agrawal et al., 2016; Goyal et al., 2017; Agrawal et al., 2018; Cadène et al., 2019; Hessel & Lee, 2020). Since these are both symptoms of multimodal neural networks failing to utilize all modalities, we treat them as the same problem, and refer to it as *modality underutilization.*

Existing work has addressed this issue by ensuring that there are no systematic biases in dataset creation (Goyal et al., 2017; Poliak et al., 2018; Suhr et al., 2019; Hudson & Manning, 2019), or that our neural network architectures and optimization algorithms are able to learn modality interactions (Hazirbas et al., 2017; Zeng et al., 2019; Song et al., 2020; Valada et al., 2020; Wang et al., 2020a;b; Wu et al., 2022). We demonstrate that even under these favorable conditions, modality underutilization can still occur in the small data regime. This has important implications for practitioners of multimodal learning. Even if they carefully design their data collection procedure to be free of biases, and use proven neural network architectures and optimization methods, they are still vulnerable to modality underutilization if they do not collect enough data.

We explain this phenomenon using a concept that we call *incidental correlation.* Incidental correlation is a spurious correlation that emerges in small datasets, despite not being a part of the underlying data generating process (DGP). For example, suppose the ground-truth DGP consists of two independent Bernoulli random variables. If we sample a small dataset from this DGP and compute the correlation between the variables, it is likely to be significantly different from zero. In other words, the dataset has higher likelihood under the alternative DGP than under the ground-truth DGP that it was sampled from.

To connect this idea to modality underutilization, we put forth an idealized DGP under which modality underutilization cannot occur. We then explain how this ideal scenario breaks down in the small data regime. The DGP is a causal graphical model (Pearl, 2009), since we consider the causal relationships between the variables. There is an unobserved variable $\mathbf{u}$ that gives rise to two observable input modalities $\mathbf{x}$ and $\mathbf{x}'$. These input modalities then cause the target $\mathbf{y}$. We refer to this DGP as the *multimodal generating process* (MGP), and show its graph in Fig. 1a. Under the MGP, we have

$$p(\mathbf{y} \mid do(\mathbf{x}), do(\mathbf{x}')) = p(\mathbf{y} \mid \mathbf{x}, \mathbf{x}'). \tag{1}$$

In causal language, $p(\mathbf{y} \mid do(\mathbf{x}), do(\mathbf{x}'))$ is identifiable, meaning that it is obtainable by estimating $p(\mathbf{y} \mid \mathbf{x}, \mathbf{x}')$ from observational data. Therefore, if we estimate $p(\mathbf{y} \mid \mathbf{x}, \mathbf{x}')$ and use it to predict the target given the inputs, the prediction is purely based on the causal paths $\mathbf{x} \to \mathbf{y}$ and $\mathbf{x}' \to \mathbf{y}$. Such a model successfully utilizes all modalities.

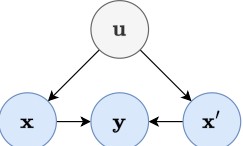

(a) Multimodal generating process (MGP). All paths between the inputs and target are causal.

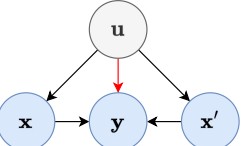

(b) Spurious generating process (SGP). The edge from $\mathbf{u}$ to $\mathbf{y}$ unblocks spurious paths between the inputs and target, which lead to modality underutilization.

Figure 1

We argue that modality underutilization occurs when this ideal scenario fails to materialize. Suppose we sample a small dataset from the MGP, where $\mathbf{u}$ and $\mathbf{y}$ are conditionally independent given the inputs. Due

to incidental correlation, our dataset can have higher likelihood under an alternative DGP with an edge from $\mathbf{u}$ to $\mathbf{y}$. This additional edge makes $\mathbf{u}$ and $\mathbf{y}$ conditionally dependent given the inputs. We call this alternative DGP the *spurious generating process* (SGP), and show its graph in Fig. 1b. The edge from $\mathbf{u}$ to $\mathbf{y}$ unblocks the spurious paths $\mathbf{x} \leftarrow \mathbf{u} \rightarrow \mathbf{y}$ and $\mathbf{x}' \leftarrow \mathbf{u} \rightarrow \mathbf{y}$, and prevents Eq. 1 from holding. Consequently, if we estimate $p(\mathbf{y} \mid \mathbf{x}, \mathbf{x}')$ and use it to predict the target given the inputs, we can no longer tell whether the predictions are based on the causal paths, or the spurious ones. This ambiguity can lead to modality underutilization. For example, a model can rely solely on the spurious path $\mathbf{x} \leftarrow \mathbf{u} \rightarrow \mathbf{y}$ for prediction, thus failing to utilize the $\mathbf{x}'$ modality.

It is beneficial to detect incidental correlation, given that it leads to modality underutilization. The main challenge in doing so is that it involves the unobserved variable $\mathbf{u}$. We circumvent this difficulty by using latent variable modeling. We specify an identifiable variational autoencoder (VAE) (Kingma & Welling, 2014) such that the relationship between the latent variable $\mathbf{z}$ and the inputs captures the incidental correlation between $\mathbf{u}$ and the target. This allows us to interpret the Kullback-Leibler (KL) divergence between the latent posterior and prior as the severity of incidental correlation. We elaborate on this procedure in Sec. 4. Since we derive our conclusions from the latent posterior, it is important to prevent learning an arbitrary latent space. We therefore use a VAE with a mixture prior and a piecewise affine decoder to achieve identifiability (Kivva et al., 2022). In order to assess its importance, we compare these results to those derived using a vanilla VAE (Kingma & Welling, 2014), which is unidentifiable. Our ablation study demonstrates that identifiability is necessary for drawing correct conclusions.

We conduct experiments to empirically verify our main claim, which is that even if the underlying DGP is free of systematic biases, incidental correlation emerges in the small data regime, and leads to modality underutilization. Our experiments consist of a toy problem with synthetic data and small neural networks, as well as realistic settings with VQA v2.0 (Goyal et al., 2017) and NLVR2 (Suhr et al., 2019), and a state-of-the-art model called FIBER (Dou et al., 2022). In all cases, our experiments consist of two stages.

The first stage shows that even when incidental correlation is absent in the large data regime, it emerges as we reduce the size of our datasets. We do this by training a VAE, and showing that the KL divergence between the latent posterior and prior is approximately zero for large datasets, and increases as the datasets become smaller. This occurs because the latent variable encodes spurious correlations between the inputs and target that are present in small datasets, and lead the latent posterior to diverge from the prior.

The second stage demonstrates that the emergence of incidental correlation leads to modality underutilization. To show this, we train a multimodal neural network and an ensemble of unimodal neural networks, and use their generalization gap as a proxy for modality underutilization. The multimodal neural network generalizes better than the unimodal ensemble in the large data regime, but this difference erodes as the datasets become smaller. We attribute this to spurious paths between the inputs and target that fail to generalize.

Practitioners of multimodal learning can use our method to detect whether incidental correlation is present in their datasets, and determine whether they should collect more data. This is useful in realistic situations, when practitioners encounter modality underutilization in data-constrained settings. While our experiments involve a large dataset, this was for the purpose of experimental control. We showed that even when incidental correlation is absent in the large data regime, it emerges when we reduce the size of our datasets. Data is less abundant in many real-world applications, and this is where our method provides value.

## 2 Related work

### 2.1 Modality underutilization

Modality underutilization is a long-standing open problem, and existing work has generally addressed it from two different directions. The first direction is to ensure that there are no systematic bias in dataset creation. This is motivated by the fact that many benchmark datasets intended to require multimodal reasoning could unwittingly be solved with a subset of modalities. Poliak et al. (2018) showed in natural language inference that tasks designed to require the use of a hypothesis and a context could be solved with hypothesis-only

baselines. A similar problem occurs in visual question answering (VQA) (Antol et al., 2015). Instead of answering a question about a specific image, VQA models often predict the same answer for a given question across a wide variety of images (Agrawal et al., 2016; Jabri et al., 2016; Agrawal et al., 2018; Cadène et al., 2019). To counter this problem, many multimodal benchmark datasets have been proposed that are carefully constructed to be free of systematic biases (Goyal et al., 2017; Hudson & Manning, 2019). We show that this precaution is not enough, and that incidental correlation can be detected in such datasets in the small data regime.

The second direction is to ensure that our neural network architectures and optimization algorithms are capable of learning modality interactions. In terms of neural network architectures, the majority take into account the idiosyncrasies of the specific modalities and prediction tasks (Hazirbas et al., 2017; Zeng et al., 2019; Song et al., 2020; Valada et al., 2020; Wang et al., 2020b), but generic approaches also exist (Jean & Cho, 2019; Baevski et al., 2022). Regarding optimization algorithms, Wang et al. (2020a) proposed to mitigate modality underutilization by averaging gradients across modalities, while Wu et al. (2022); Sun et al. (2021) took the approach of balancing the modality-specific learning rates. Meanwhile, Gat et al. (2020) introduced a regularization term based on functional entropy to balance the contributions of each modality.

## 2.2 Modeling unobserved confounding

Our work is related to causal inference approaches that address unobserved confounding between multiple treatments and a single outcome (Wang et al., 2017; Frot et al., 2018; Tran & Blei, 2018; Heckerman, 2018; Janzing & Schölkopf, 2018; Wang & Blei, 2019; Ranganath & Perotte, 2018; D'Amour, 2019; Ćevid et al., 2020; Puli et al., 2020; Agrawal et al., 2021; Wang & Blei, 2021). These studies leveraged the correlations among the multiple treatments as observable signatures for inferring the shared unobserved confounder. We also draw inspiration from Makino et al. (2022), who adjusted for unobserved confounding in the context of multitask learning, where there is a single input and multiple targets.

## 2.3 Identifiable latent variable modeling

Informally, a latent variable model is identifiable if the true joint distribution over the observed and latent variables can be learned from data. The precise definition is context-dependent, and we provide ours in the appendix. Identifiablity is important in the context of this work, since we derive our conclusions from the latent posterior. The theory originated in nonlinear ICA (Hyvärinen & Morioka, 2016; 2017; Hyvärinen et al., 2019), which was extended to VAEs in Khemakhem et al. (2020). These works assumed there is an observed auxiliary variable that the latent prior factorizes over, such as a timestamp or a class label. Since auxiliary variables are not always available, there have been efforts to relax this requirement. Wang et al. (2021) achieved this with Brenier maps and input convex neural networks, and Moran et al. (2022) did so using a sparse model in which each dimension of the observed data depends on a small subset of the latent variables. There also exist approaches based on the principle of independent mechanisms from causality (Gresele et al., 2021; Lachapelle et al., 2022). We adopt the assumptions of Kivva et al. (2022), who achieve identifiability without auxiliary information via a mixture prior and a piecewise affine decoder. This approach is appealing to us because it is consistent with conventional modeling choices that scale to high-dimensional data (Jiang et al., 2017; Falck et al., 2021). It has also been shown to perform well in terms of various empirical measures of identifiability (Willetts & Paige, 2021). The topic of identifiability is also related to disentanglement, which can be thought of as identifiability up to permutation and component-wise transformation (Higgins et al., 2017; Li & Mandt, 2018; Locatello et al., 2019; Bengio et al., 2020; Bai et al., 2021; Lachapelle et al., 2022). Our work also joins a direction of research that demonstrates the practical utility of identifiability. While this topic has been relatively unexplored (Hyvärinen et al., 2023), Lopez et al. (2023) demonstrated its importance for modeling single-cell genomics data.

# 3 Incidental correlation

We explain modality underutilization using a concept that we call incidental correlation. We define this as a spurious correlation that emerges in the small data regime, despite not being a part of the underlying DGP. Small datasets that are sampled from the MGP, where $\mathbf{u}$ and $\mathbf{y}$ are conditionally independent given the inputs, can have higher likelihood under the SGP, where this independence does not hold. In the SGP, the correlation between the inputs and targets flow through the causal paths $\mathbf{x} \to \mathbf{y}$ and $\mathbf{x}' \to \mathbf{y}$, as well as the spurious paths $\mathbf{x} \leftarrow \mathbf{u} \to \mathbf{y}$ and $\mathbf{x}' \leftarrow \mathbf{u} \to \mathbf{y}$. As we discussed in Sec. 1, these spurious paths contribute to modality underutilization.

The severity of this effect depends on the degree to which $\mathbf{u}$ and $\mathbf{y}$ are conditionally dependent given the inputs. In order to explain this, let us define $\mathbf{y}$ in the SGP as

$$\mathbf{y} := f_{\mathbf{y}}(\mathbf{x}, \mathbf{x}', g(\mathbf{u}), \epsilon_{\mathbf{y}}), \tag{2}$$

where $\epsilon_{\mathbf{y}}$ is exogenous noise, and $g(\mathbf{u})$ is a function that determines the degree to which $\mathbf{u}$ and $\mathbf{y}$ are conditionally dependent given the inputs. On one extreme, if $g(\mathbf{u})$ is a constant function, $\mathbf{u}$ and $\mathbf{y}$ are conditionally independent given the inputs. This makes the SGP equivalent to the MGP. On the other extreme, if $g(\mathbf{u}) = \mathbf{y}$, $\mathbf{u}$ and $\mathbf{y}$ are perfectly dependent given the inputs. In this case, $\mathbf{u}$ and $\mathbf{y}$ collapse into a single variable, which we refer to as $\mathbf{y}$. In the corresponding graph shown in Fig. 2a, the edges between the inputs and target are bidirectional, forming a Markov random field. The data distribution is given by

$$p(\mathbf{x}, \mathbf{x}', \mathbf{y}) \propto \exp(\phi_{\mathbf{x}}(\mathbf{x}) + \phi_{\mathbf{x}'}(\mathbf{x}') + \phi_{\mathbf{y}}(\mathbf{y}) + \phi_{\mathbf{x},\mathbf{y}}(\mathbf{x}, \mathbf{y}) + \phi_{\mathbf{x}',\mathbf{y}}(\mathbf{x}', \mathbf{y})),$$

where the $\phi$'s are potential functions. Under this distribution, the prediction for $\mathbf{y}$ is given by

$$\arg\max_{\mathbf{y}} \phi_{\mathbf{y}}(\mathbf{y}) + \phi_{\mathbf{x},\mathbf{y}}(\mathbf{x}, \mathbf{y}) + \phi_{\mathbf{x}',\mathbf{y}}(\mathbf{x}', \mathbf{y}).$$

If $\phi_{\mathbf{y}}(\mathbf{y})$ is uniform, and thus constant with respect to $\mathbf{y}$, this corresponds to an ensemble of the two unimodal neural networks $\phi_{\mathbf{x},\mathbf{y}}(\mathbf{x}, \mathbf{y})$ and $\phi_{\mathbf{x}',\mathbf{y}}(\mathbf{x}', \mathbf{y})$. In other words, multimodal learning is equivalent to unimodal ensembling in this particularly degenerate case of the SGP. In practice, the severity of incidental correlation is likely to fall somewhere between these two extremes.

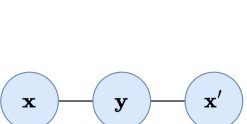

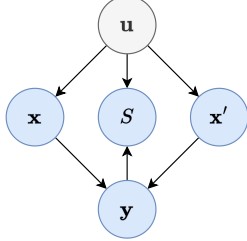

(a) When $\mathbf{u}$ and $\mathbf{y}$ collapse into a single variable which we call $\mathbf{y}$, the graph becomes a Markov random field because the edges between the inputs and target are bidirectional.

(b) Selection bias is induced by conditioning on the collider $S$, which makes $\mathbf{u}$ and $\mathbf{y}$ conditionally independent given the inputs.

Figure 2

Incidental correlation is not the only way that $\mathbf{u}$ and $\mathbf{y}$ can become conditionally dependent given the inputs. Another way this can occur is due to selection bias, where there is an additional node $S \in \{0, 1\}$ that represents a selection criteria (Bareinboim & Pearl, 2012). We show the corresponding graph in Fig. 2b. Data is sampled from a DGP, and is included in the dataset when $S = 1$. Since $S$ is a child of $\mathbf{u}$ and $\mathbf{y}$, this induces a collider bias and makes $\mathbf{u}$ and $\mathbf{y}$ conditionally dependent given the inputs. Selection bias is an example of a systematic bias, so we do not consider it in this work. While systematic biases are indeed problematic, the purpose of our work is to show that even under favorable conditions where there are no systematic biases, modality underutilization can occur in the small data regime due to incidental correlation.

# 4 Detecting incidental correlation

Suppose we have a multimodal dataset $\mathcal{D} = \{\mathbf{x}^{(n)}, \mathbf{x}'^{(n)}, \mathbf{y}^{(n)}\}_{n=1}^N$, where $N$ is the size of the dataset. It is beneficial to be able to detect whether incidental correlation is present in $\mathcal{D}$, since if it is, spurious correlations between the inputs and target can lead to modality underutilization. We propose to detect incidental correlation using latent variable modeling.

Our method is based on the following high-level idea, which we elaborate on in the following paragraphs. If there is no incidental correlation in $\mathcal{D}$, then it has highest likelihood under the MGP, and $\{\mathbf{x}, \mathbf{x}'\}$ is sufficient for predicting the target. In contrast, if incidental correlation is present, $\mathcal{D}$ has higher likelihood under the SGP, and $\{\mathbf{x}, \mathbf{x}'\}$ is no longer sufficient for predicting the target. We therefore introduce a latent variable $\mathbf{z}$, and predict the target based on $\{\mathbf{x}, \mathbf{x}', \mathbf{z}\}$. $\mathbf{z}$ encodes any additional information that is useful for predicting the target, given the inputs. This additional information corresponds to the spurious correlations between the inputs and target.

To achieve this, we specify a conditional VAE (Kingma & Welling, 2014; Kingma et al., 2014; Sohn et al., 2015) with an encoder $q(\mathbf{z} \mid \mathbf{x}, \mathbf{x}', \mathbf{y})$ and a decoder $p(\mathbf{y} \mid \mathbf{x}, \mathbf{x}', \mathbf{z})$, and maximize the lower bound of $\log p(\mathbf{y} \mid \mathbf{x}, \mathbf{x}')$, which is

$$\mathbb{E}_{q(\mathbf{z}|\mathbf{x},\mathbf{x}',\mathbf{y})}[\log p(\mathbf{y} \mid \mathbf{x}, \mathbf{x}', \mathbf{z})] - D_{KL}(q(\mathbf{z} \mid \mathbf{x}, \mathbf{x}', \mathbf{y}) \parallel p(\mathbf{z})). \tag{3}$$

This is called the evidence lower-bound (ELBO), and is derived in the appendix. The KL term in the ELBO acts as a regularizer, and discourages putting information in $\mathbf{z}$ unless necessary. This means that if there are no spurious correlations between the inputs and targets, the encoder does not need to do anything, and the KL term is driven to zero. However, if incidental correlation is present in $\mathcal{D}$, then the spurious correlations between the inputs and target are represented by $\mathbf{z}$, and the KL term increases. We can therefore interpret the KL divergence between the latent posterior and prior as the severity of incidental correlation. Another way to interpret this is as a conditional independence test. We are using the KL term to determine whether $\mathbf{u}$ and $\mathbf{y}$ are conditionally independent given the inputs, without having to observe $\mathbf{u}$.

If the KL divergence between the latent posterior and prior is close to zero, this leads us to conclude that incidental correlation is absent from $\mathcal{D}$. In order to prevent false negatives, we need to take precautions to avoid a common failure mode of VAEs called posterior collapse. Posterior collapse refers to the latent posterior being independent of the data, and equaling the prior. This has often been attributed to optimization issues specific to VAEs (Dieng et al., 2019; Lucas et al., 2019; Razavi et al., 2019), but was linked to latent variable identifiability by Wang et al. (2021). Posterior collapse is not possible when the true model does not exhibit posterior collapse, and is identifiable up to an invertible affine transformation. In the context of an observed variable $\mathbf{x}$ and a latent variable $\mathbf{z}$, Wang et al. (2021) defined unidentifiability as $p(\mathbf{x} \mid \mathbf{z}) = p(\mathbf{x})$, and posterior collapse as $p(\mathbf{z} \mid \mathbf{x}) = p(\mathbf{z})$. The authors showed that the former implies the latter. Conversely, if the true model satisfies $p(\mathbf{x} \mid \mathbf{z}) \neq p(\mathbf{x})$, then its posterior does not collapse. If we estimate the true model up to an invertible affine transformation, then $p(\mathbf{x} \mid \mathbf{z}) \neq p(\mathbf{x})$ holds for our model, and its posterior does not collapse. In other words, a non-collapsed posterior cannot be transformed into a collapsed posterior by an invertible affine transformation.

Therefore, we adopt the assumptions of Kivva et al. (2022) to achieve identifiability up to an invertible affine transformation. Informally, the two assumptions are that that the latent prior is a Gaussian mixture, and that the decoder is piecewise affine. In the appendix, we state the relevant assumptions and identifiability results of Kivva et al. (2022). The latent prior is given by

$$p(\mathbf{z}) = \sum_{k=1}^{K} P(C = k)\mathcal{N}(\mathbf{z}; \mu_c, \Sigma_c),$$

where $K$ is the number of components, and $C$ is a categorical variable. The corresponding graph is shown in Fig. 3.

A key requirement in specifying our model is that $\mathbf{z}$ and $\mathbf{y}$ are conditionally dependent given $\{\mathbf{x}, \mathbf{x}'\}$, since this is what allows us to model incidental correlation with the latent posterior. This can also be achieved

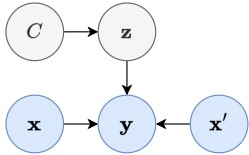

Figure 3: Our VAE has a mixture prior over $\mathbf{z}$, where $C$ is the mixture component.

with an alternative model with edges from $\mathbf{z}$ to $\{\mathbf{x}, \mathbf{x}'\}$. These edges are present in the SGP, as seen in Fig. 1b. However, it is unnecessary to include these edges in our model, since $\{\mathbf{x}, \mathbf{x}'\}$ are always conditioned on. The only unblocked path from $\mathbf{z}$ to $\mathbf{y}$ conditioned on $\{\mathbf{x}, \mathbf{x}'\}$ is the edge from $\mathbf{z}$ to $\mathbf{y}$, so we chose to make $\mathbf{z}$ to $\{\mathbf{x}, \mathbf{x}'\}$ marginally independent in our model. This has the practical benefit of simplifying the implementation of the KL divergence between the latent posterior and prior.

To summarize, we train an identifiable VAE in order to detect incidental correlation, where the latent posterior encodes the spurious correlations between the inputs and target. This allows us to interpret the KL divergence between the latent posterior and prior as the severity of incidental correlation. This value should be zero when datasets have higher likelihood under the MGP relative to the SGP, since the inputs are sufficient for modeling the target. However, this value will be positive if, due to incidental correlation, there is additional spurious information that is predictive of the target.

There are three conditions under which we are correct to interpret the KL divergence between the latent posterior and prior as the severity of incidental correlation. First, if our causal graph in Fig. 1b correctly describes the true DGP, then any information in addition to $\{\mathbf{x}, \mathbf{x}'\}$ that is useful for predicting $\mathbf{y}$ is spurious. Second, if the VAE is identifiable, the latent posterior represents this spurious information. Third, if the VAE is trained until convergence, the KL term is minimized, meaning that it is only positive when the spurious information is present.

## 5 Experiments

### 5.1 Toy problem

We begin by verifying our claims with a toy problem with synthetic data. Our toy problem is designed so that in the large data regime, multimodal learning successfully utilizes all modalities. We then reduce the size of the dataset, while keeping everything else the same. This gradually induces incidental correlation, which subsequently leads to modality underutilization.

The prediction task is inspired by the exclusive or (XOR) operation. XOR is a logical connective that outputs true if exactly one of two input conditions is true. It is relevant from the perspective of comparing multimodal and unimodal learning, because knowing only one of the two input conditions bears no information on the output value. The DGP is

$$\begin{aligned}
\mathbf{u} &\sim \mathcal{N}(\mathbf{0}, I), \\
\mathbf{x} &\sim \mathcal{N}(\mathbf{u}_J + b \cdot \mathbf{1}, \sigma^2 I), \\
\mathbf{x}' &\sim \mathcal{N}(\mathbf{u}_{-J}, \sigma^2 I), \\
Y &:= \mathrm{Ber}(\sigma(100\mathbf{x}^\top \mathbf{x}')),
\end{aligned}$$

where $\mathbf{u} \in \mathbb{R}^{2D}$, $\mathbf{x}, \mathbf{x}' \in \mathbb{R}^D$, and $Y \in \{0, 1\}$. Here, we set $b = 1.5$ and $\sigma = 0.1$, and provide additional results with $b = 3$ and $\sigma = 1$ in the appendix.

The mean of $\mathbf{x}$ is a random subset of $\mathbf{u}$ that is offset by $b \cdot \mathbf{1}$. This offset serves an important purpose, which we elaborate on in the next paragraph. The set of indices $J$ are a randomly sampled half of the full set of indices $\{1, \ldots, 2D\}$. We denote the remaining half of indices as $-J = \{1, \ldots, 2D\} \setminus J$. This DGP is an MGP, since the target is defined without involving $\mathbf{u}$. Since its definition includes an interaction between all

modalities, classifying the target requires using all modalities. However, as we will show, a spurious solution emerges in the small data regime due to the randomness of sampling.

In order to explain this visually, we illustrate the prediction task in Fig. 4 in the simplest setting with scalar inputs, i.e. $D = 1$. As seen in Fig. 4a, in the large data regime, a multimodal neural network can learn to predict $Y = 1$ when the inputs have the same sign, and $Y = 0$ otherwise. In contrast, a unimodal ensemble cannot express this decision rule because it considers each input separately. Therefore, multimodal learning generalizes better than unimodal learning in the large data regime. Fig. 4b shows that the situation changes in the small data regime. Since we offset the mean of $X$ by $b = 1.5$, a unimodal neural network can accurately classify $Y$ while only looking at $X'$. Therefore, the predictive advantage of multimodal learning over unimodal learning deteriorates in the small data regime.

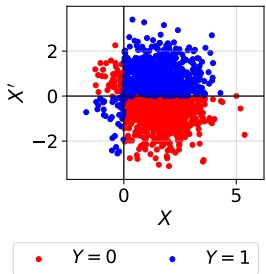 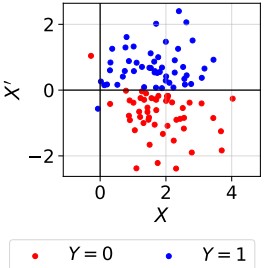

(a) In the large data regime, both modalities are required in order to classify the target.

(b) In the small data regime, the target can be classified using a single modality.

Figure 4: Toy problem with scalar inputs.

In the following experiments, we sample several datasets from the DGP that vary only in the number of examples $N \in \{100, 400, 1600, 6400, 25600\}$. We then randomly select 60% as the training set, 20% as the validation set, and the remaining 20% as the test set. We train for a certain number of epochs $M$ that depends on $N$, and early stop when the validation loss does not improve for $\lfloor 0.1 \cdot M \rfloor$ epochs. A table with the values of $M$ and $N$ is in the appendix. We use a batch size of 128 for all experiments. We show the results for $D = 1$ here, and those for $D = 16$ in the appendix. For each $N$, we train ten models starting from different randomly-initialized parameters, and report the mean and standard deviation.

### 5.1.1 Detecting incidental correlation with u observed

We detect incidental correlation in two ways: using a simple heuristic with $\mathbf{u}$ observed, and with our VAE-based detection method described in Sec. 4, where $\mathbf{u}$ is unobserved. In both cases, we show that incidental correlation emerges in the small data regime. The following heuristic with $\mathbf{u}$ observed serves as a sanity check, since if we cannot detect incidental correlation with $\mathbf{u}$ observed, we should not be able to detect it when it is unobserved. We define an alternative DGP where $Y$ is redefined as

$$Y := \text{Ber}(\sigma(\tau \mathbf{x}^\top \mathbf{x}' + \alpha^\top \mathbf{u})).$$

This alternative DGP is a SGP, since there is an edge from $\mathbf{u}$ to $Y$. We then compute

$$\arg\max_\alpha \sum_{n=1}^N \log P(y^{(n)} \mid \mathbf{x}^{(n)}, \mathbf{x}'^{(n)}, \alpha^\top \mathbf{u}^{(n)})$$

on the training set, and report the value of $\|\alpha\|$ corresponding to the maximum log-likelihood on the validation set. The only trainable parameter is $\alpha \in \mathbb{R}^{2D}$, which we optimize using the Adam optimizer (Kingma & Ba, 2015) with a learning rate of $10^{-3}$.

Since $\mathcal{D}$ was sampled from an MGP, where $\mathbf{u}$ and $Y$ are conditionally independent given the inputs, $\alpha^\top \mathbf{u}$ should play no part in maximizing the likelihood. Therefore, we can interpret $\|\alpha\|$ as the degree of incidental

correlation. The results are shown in Fig. 5a. As expected, $\|\alpha\|$ is centered near zero in the large data regime, and increases as the dataset becomes smaller. This confirms that incidental correlation emerges in the small data regime.

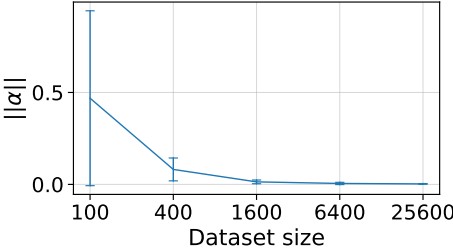
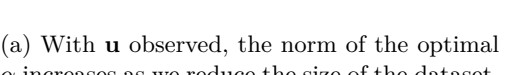
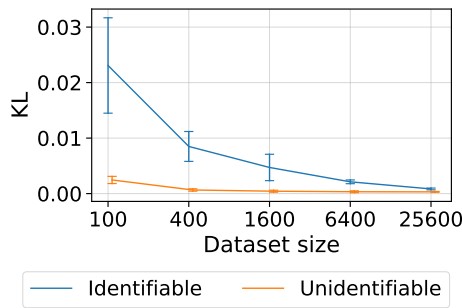

(a) With **u** observed, the norm of the optimal $\alpha$ increases as we reduce the size of the dataset.

(b) With **u** unobserved, the KL divergence between the latent posterior and prior increases as we reduce the size of the dataset. This pattern is more discernible with the identifiable VAE.

Figure 5: Detecting incidental correlation on the toy problem with $D = 1$, $b = 1.5$, and $\sigma = 0.1$. Reducing the size of the dataset makes incidental correlation emerge, both when **u** is observed and unobserved.

### 5.1.2 Detecting incidental correlation with u unobserved

Given the success of our sanity check with **u** observed, we move on to detecting incidental correlation with our VAE-based method, where **u** is unobserved. Our VAE consists of an encoder and a decoder, which are both implemented using MLPs with two hidden layers with 128 neurons each, and ReLU activations. This choice of activation function in the decoder satisfies an assumption for achieving identifiability (Kivva et al., 2022). We additionally train a vanilla VAE (Kingma & Welling, 2014) with the same hyperparameters except for sigmoid activations in the decoder, which violates the assumptions of Kivva et al. (2022). We refer to this as the unidentifiable VAE. This ablation study allows us to check whether identifiability helps us draw correct conclusions based on the latent posterior.

The two VAEs have the same encoder, and their decoder is also the same except for the choice of activation function. Their biggest difference is in their prior over the latent variable $\mathbf{z} \in \mathbb{R}^{16}$. For the encoder, two MLPs each take as input the concatenation of $\{\mathbf{x}, \mathbf{x}', \mathbf{y}\}$ and return the mean and covariance parameters of the multivariate Gaussian distribution $q(\mathbf{z} \mid \mathbf{x}, \mathbf{x}', \mathbf{y})$. For the decoder, an MLP takes as input the concatenation of $\{\mathbf{x}, \mathbf{x}', \mathbf{z}\}$ and outputs the values of the categorical distribution $p(\mathbf{y} \mid \mathbf{x}, \mathbf{x}', \mathbf{z})$. As for the prior, in our identifiable VAE, we use a Gaussian mixture prior to satisfy an assumption from Kivva et al. (2022). There are 16 mixture components, where each component is Gaussian with diagonal covariance. The mixture components are uniformly initialized, and the Gaussian parameters are Xavier normal initialized. All parameters of the mixture prior are updated during training. The prior in our unidentifiable VAE is Gaussian with zero mean and unit covariance. The parameters of the prior are updated during training for the identifiable VAE, but not for the unidentifiable VAE. Both VAEs are trained with the Adam optimizer (Kingma & Ba, 2015) with a learning rate of $10^{-4}$.

Our results in Fig. 5b show that for the identifiable VAE, the KL divergence between the latent posterior and prior is centered near zero in the large data regime, but increases as the dataset becomes smaller. This means that incidental correlation emerges in the small data regime, as we expected. This relationship does not hold for the unidentifiable VAE, whose posterior collapses. This supports the theory of Kivva et al. (2022), and confirms the importance of achieving identifiability when drawing conclusions based on the latent posterior.

### 5.1.3 Comparing multimodal and unimodal learning

Having detected incidental correlation, we move on to comparing multimodal and unimodal learning in its presence. Our multimodal neural network is implemented using an MLP which takes as input the concatenation of $\{\mathbf{x}, \mathbf{x}'\}$, and outputs the values of the Bernoulli distribution $p(\mathbf{y} \mid \mathbf{x}, \mathbf{x}')$. The ensemble of unimodal neural networks consists of two MLPs. The MLPs separately take as input $\mathbf{x}$ and $\mathbf{x}'$, and output the values of $p(\mathbf{y} \mid \mathbf{x})$ and $p(\mathbf{y} \mid \mathbf{x}')$. These unimodal predictions are averaged to form the ensemble prediction. All of these MLPs have two hidden layers with 128 neurons each, and ReLU activations. They are trained with the Adam optimizer (Kingma & Ba, 2015) with a learning rate of $10^{-3}$.

We interpret the difference in log-probability of the target given the inputs between the multimodal neural network and the unimodal ensemble as a proxy for modality underutilization. We refer to this difference as the generalization gap, and use it as the vertical axis in our results in Fig. 6. Our results show that the multimodal neural network generalizes better than the unimodal ensemble in the large data regime, but this difference erodes when datasets become smaller. This serves as strong evidence that incidental correlation contributes to modality underutilization.

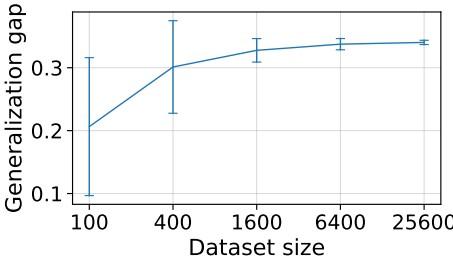

Figure 6: Comparing multimodal and unimodal learning on the toy problem with $D = 1$, $b = 1.5$, and $\sigma = 0.1$. Multimodal learning generalizes better than unimodal learning in the large data regime, but the difference narrows as we reduce the size of the dataset.

## 5.2 VQA v2.0 and NLVR2

Having verified our claims on the toy problem, we move on to realistic settings with two popular benchmark visual language datasets called VQA v2.0 (Goyal et al., 2017) and NLVR2 (Suhr et al., 2019), and a state-of-the-art neural network architecture called FIBER (Dou et al., 2022). Both of these datasets are designed to be less systematically biased than their respective first iterations. Therefore, they represent ideal test beds for us to demonstrate that incidental correlation can occur in the absence of systematic dataset biases.

Similar to our experiments with the toy problem, our experiments in this section consist of two stages, where we first detect the presence of incidental correlation, and then compare the generalization gap between multimodal and unimodal learning.

**VQA v2.0**  We experiment with VQA v2.0 (Goyal et al., 2017), which is a revised version of VQA v1.0 (Antol et al., 2015) that is constructed to be free of systematic biases. Models trained on VQA v1.0 were frequently observed to ignore the image when answering a question. This is thought to occur because the dataset exhibits strong correlations between the questions and answers. To remedy this, the authors of VQA v2.0 ensured that each question in the dataset is associated with a pair of similar images that result in two different answers.

The dataset consists of images and questions as the inputs, and answers as the target. Each target is a vector representing the 3,129 most common answers in the training and validation sets. Each element of this vector represents an answer, and the values are in $[0, 1]$ to reflect the labelers' uncertainty. The log-probability of the target is the sum of log-probabilities of the individual answers. This reflects standard practice when working

with this dataset. We construct several versions of the dataset that vary only in their number of examples. To do so, we first merge the training and validation set to use as the full set of examples. We do not include the test set, because the ground-truth labels are not publicly available for it. The creators of the dataset indexed these examples in terms of unique images, where each image is associated with a variable-sized set of questions and answers. To sample a dataset, we randomly sample $N \in \{500, 2000, 8000, 32000, 128000\}$ unique images from the full set of 217,522, and split them into training, validation, and test sets, using a 60%, 20%, and 20% split. Thus, the dataset sampling procedure is consistent across all choices of $N$. Similarly to the toy problem, in the following experiments we train for a certain number of epochs $M$ for a given $N$, whose value are provided in the appendix. In all cases, we early stop when the validation loss stops improving for 20 epochs. We use a batch size of 512 for all experiments. For each $N$, we train five models from different random initializations, and report the mean and standard deviation.

**NLVR2**   We additionally experiment with NLVR2 (Suhr et al., 2019), which is designed to be less systematically biased than NLVR (Suhr et al., 2017), its previous iteration. NLVR2 represents the binary classification task of answering a true or false question regarding a pair of images. Therefore, there are three inputs, where two are images and one is text. We merge the training, validation, and test examples together, for a total of 59,677. We then randomly sample $N \in \{200, 800, 3200, 12800, 51200\}$ examples from the full set, and split them into training, validation, and test sets again using a 60%, 20%, and 20% split.

For both datasets, instead of processing the raw image and text inputs, we instead use the embeddings produced by a state-of-the-art neural network called FIBER (Dou et al., 2022) that was pretrained on a diverse set of datasets and tasks. We use a version of the model that was released by the authors that uses a Swin Transformer as the image backbone (Liu et al., 2021), and RoBERTa as the text backbone (Liu et al., 2019). The image and text embeddings are both in $\mathbb{R}^{768}$. For VQA v2.0, we denote the image embedding as $\mathbf{x}$, and the text embedding as $\mathbf{x}'$. For NLVR2, we denote the two image embeddings as $\mathbf{x}_1$ and $\mathbf{x}_2$, and the text embedding as $\mathbf{x}'$. During training, we freeze the weights of the image and text backbones that produce the embeddings.

### 5.2.1   Detecting incidental correlation

In order to detect incidental correlation, we train identifiable and unidentifiable VAEs, and report the KL divergence between the latent posterior and prior on the test set, using the weights that minimize the validation loss. The VAEs have encoders and decoders that are both implemented using MLPs with two hidden layers with 512 neurons each. The MLPs are parameterized with ReLU activations, except for the decoder of the unidentifiable VAE, which uses sigmoid activations. The choice of sigmoid activations leads the unidentifiable VAE to violate the piecewise affine decoder assumption in Kivva et al. (2022). For both VAEs, the latent variable $\mathbf{z}$ is in $\mathbb{R}^{512}$. The identifiable VAE has a mixture prior with 128 components, where each component is Gaussian with diagonal covariance. Both the categorical distribution over the components, and the mean and diagonal entries of the components are randomly initialized and updated during training. The components are uniform initialized, and the Gaussian parameters are Xavier normal initialized. The prior for the unidentifiable VAE is Gaussian with mean zero and unit covariance, and its parameters are not updated during training. Both VAEs are optimized with the Adam optimizer (Kingma & Ba, 2015) with a learning rate of $10^{-4}$.

In Fig. 7, we show our results for detecting incidental correlation in VQA v2.0 and NLVR2. The conclusions are similar to what we saw for the toy problem. With the identifiable VAE, the KL divergence between the latent posterior and prior is close to zero in the large data regime. This implies that the dataset is consistent with the MGP, where there are no spurious paths between the inputs and targets. However, the KL term increases as we reduce the size of the dataset. This is because small datasets have higher likelihood under the SGP, where there are spurious paths between the inputs and target. These spurious correlations are encoded into the latent posterior, making it diverge from the prior.

In contrast, the results with the unidentifiable VAE are unclear. For both VQA v2.0 and NLVR2, the posterior of the unidentifiable VAE collapses when it does not for its identifiable counterpart. Also, the non-monotonicity of the KL term with respect to dataset size makes detecting incidental correlation difficult.

Similar to the toy problem, these results support the theory of Kivva et al. (2022), and demonstrate that identifiability is critical when drawing conclusions based on the latent posterior.

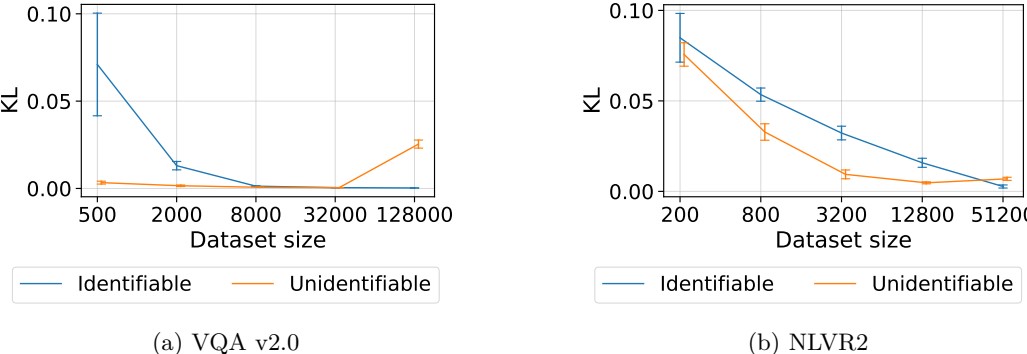

(a) VQA v2.0                                (b) NLVR2

Figure 7: Results on VQA v2.0 and NLVR2 for detecting incidental correlation. Incidental correlation emerges in the small data regime, and this pattern is more discernible with the identifiable VAE.

### 5.2.2 Comparing multimodal and unimodal learning

We now compare how a multimodal neural network generalizes relative to a unimodal ensemble. This generalization gap is a proxy for modality underutilization. For both datasets, the multimodal and unimodal neural networks are implemented as MLPs with two hidden layers with 512 units each, and ReLU activations, which are trained with the Adam optimizer (Kingma & Ba, 2015) with a learning rate of $10^{-4}$.

For VQA v2.0, the multimodal neural network takes in the concatenation of $\{\mathbf{x}, \mathbf{x}'\}$ and outputs $\log p(\mathbf{y} \mid \mathbf{x}, \mathbf{x}')$. Since there are three inputs in NLVR2, the multimodal neural network takes in the concatenation of $\{\mathbf{x}_1, \mathbf{x}_2, \mathbf{x}'\}$ and outputs $\log p(\mathbf{y} \mid \mathbf{x}_1, \mathbf{x}_2, \mathbf{x}')$. The unimodal ensemble for VQA v2.0 consists of two MLPs that separate take in $\mathbf{x}$ and $\mathbf{x}'$, and output $p(\mathbf{y} \mid \mathbf{x})$ and $p(\mathbf{y} \mid \mathbf{x}')$. Similarly, for NLVR2, three MLPs take in $\mathbf{x}_1$, $\mathbf{x}_2$ and $\mathbf{x}'$, and output $p(\mathbf{y} \mid \mathbf{x}_1)$, $p(\mathbf{y} \mid \mathbf{x}_2)$, and $p(\mathbf{y} \mid \mathbf{x}')$. In both cases, the ensemble output is the log of the average of the unimodal outputs.

Our results in Fig. 8 show that for both VQA v2.0 and NLVR2, the multimodal neural network generalizes better than the unimodal ensemble in the large data regime. This represents the ideal setting where multimodal learning works as intended. However, as datasets become smaller, multimodal learning no longer generalizes better than the unimodal ensemble. This is because incidental correlation makes the inputs and target spuriously correlated, and the multimodal neural network learns to predict the target using these spurious paths.

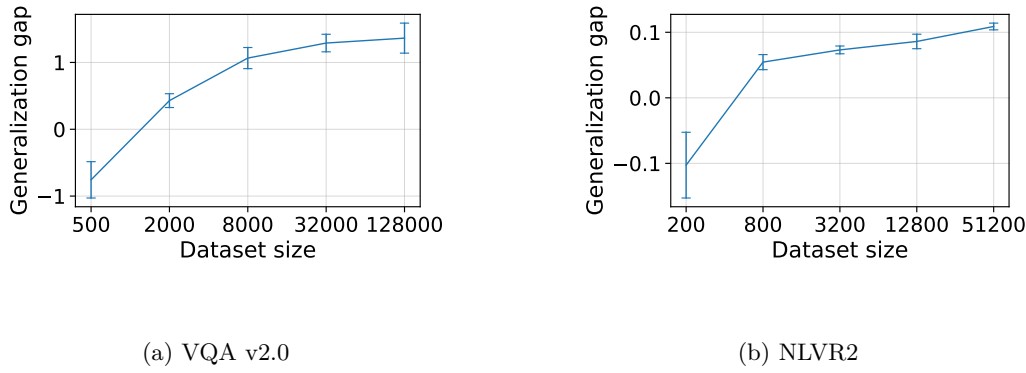

(a) VQA v2.0                                (b) NLVR2

Figure 8: Results on VQA v2.0 and NLVR2 for comparing multimodal and unimodal learning. The generalization gap between multimodal and unimodal learning narrows in the small data regime.

# 6    Conclusion

We have demonstrated using both synthetic and real data that incidental correlation emerges in the small data regime, and leads to modality underutilization. The most important takeaway is that when pursuing multimodal learning, it is not enough to just worry about systematic dataset biases, or neural network architectures and optimization algorithms. Our results show that even in an ideal setting where these concerns are met, and multimodal neural networks should successfully utilize all modalities, they fail to do so in the small data regime. In addition to the existing concerns, we must also acquire a sufficient amount of data in order for multimodal learning to be successful. When data is scarce, it may be futile to try to improve multimodal learning without resolving the issue of incidental correlation. Our proposed method enables practitioners to check whether incidental correlation is present in their datasets, and determine whether they need to collect more data.

So far, we have reported the issue of incidental correlation, and shown that it is problematic for multimodal learning. However, we have not yet proposed a way to mitigate it. We believe this is a promising direction for future work, since this can improve the efficacy of multimodal learning in data-scarce settings. One example is to use our method in the acquisition step of active learning. We can collect data to reduce the KL divergence between the latent posterior and prior, which corresponds to collecting data that does not exhibit incidental correlation.

**Acknowledgments**

This work was supported by grants from the National Institutes of Health (P41EB017183), the National Science Foundation (HDR-1922658, CHE-2231174, DMS-2310831), the Gordon and Betty Moore Foundation (9683), Hyundai Motor Company (Uncertainty in Neural Sequence Modeling), Samsung Advanced Institute of Technology (Next Generation Deep Learning: From Pattern Recognition to AI), and the Office of Naval Research (N00014-23-1-2590).

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

# A Appendix

## A.1 Derivations

The ELBO is

$$
\begin{aligned}
\log p(\mathbf{y} \mid \mathbf{x}, \mathbf{x}') &= \log \int_{\mathbf{z}} p(\mathbf{y}, \mathbf{z} \mid \mathbf{x}, \mathbf{x}') \mathrm{d}\mathbf{z} \\
&= \log E_{q(\mathbf{z}\mid\mathbf{x},\mathbf{x}',\mathbf{y})} \left[ \frac{p(\mathbf{y}, \mathbf{z} \mid \mathbf{x}, \mathbf{x}')}{q(\mathbf{z} \mid \mathbf{x}, \mathbf{x}', \mathbf{y})} \right] \\
&\geq E_{q(\mathbf{z}\mid\mathbf{x},\mathbf{x}',\mathbf{y})} \left[ \log \frac{p(\mathbf{y} \mid \mathbf{x}, \mathbf{x}', \mathbf{z}) p(\mathbf{z} \mid \cancel{\mathbf{x}, \mathbf{x}'})}{q(\mathbf{z} \mid \mathbf{x}, \mathbf{x}', \mathbf{y})} \right] \\
&= E_{q(\mathbf{z}\mid\mathbf{x},\mathbf{x}',\mathbf{y})} [\log p(\mathbf{y} \mid \mathbf{x}, \mathbf{x}', \mathbf{z})] - D_{KL}(q(\mathbf{z} \mid \mathbf{x}, \mathbf{x}', \mathbf{y}) \parallel p(\mathbf{z})).
\end{aligned}
$$

## A.2 Results from Kivva et al. (2022)

We adopt the following definition of identifiability from Kivva et al. (2022). Let $f_\sharp P$ denote the pushforward measure of $P$ by $f$. Let $\mathcal{P}$ be a family of probability distributions on $\mathbb{R}^m$ and $\mathcal{F}$ be a family of functions $f : \mathbb{R}^m \to \mathbb{R}^n$.

1. For $(P, f) \in \mathcal{P} \times \mathcal{F}$ we say that the prior $P$ is *identifiable (from $f_\sharp P$) up to an affine transformation* if for any $(P', f') \in \mathcal{P} \times \mathcal{F}$ such that $f_\sharp P \equiv f'_\sharp P'$ there exists an invertible affine map $h : \mathbb{R}^m \to \mathbb{R}^m$ such that $P' = h_\sharp P$ (i.e., $P'$ is the pushforward measure of $P$ by $h$).

2. For $(P, f) \in \mathcal{P} \times \mathcal{F}$ we say that the pair $(P, f)$ is *identifiable (from $f_\sharp P$) up to an affine transformation* if for any $(P', f') \in \mathcal{P} \times \mathcal{F}$ such that $f_\sharp P \equiv f'_\sharp P'$ there exists an invertible affine map $h : \mathbb{R}^m \to \mathbb{R}^m$ such that $f' = f \circ h^{-1}$ and $P' = h_\sharp P$.

$f$ is said to be *weakly injective* if (i) there exists $x_0 \in \mathbb{R}^n$ and $\delta > 0$ s.t. $|f^{-1}(\{x\})| = 1$ for every $x \in B(x_0, \delta) \cap f(\mathbb{R}^m)$, and (ii) $\{x \in \mathbb{R}^n : |f^{-1}(\{x\})| = \infty\} \subseteq f(\mathbb{R}^m)$ has measure zero with respect to the Lebesgue measure on $f(\mathbb{R}^m)$. According to Kivva et al. (2022), ReLU networks are generally weakly injective under simple assumptions on their architecture.

Assuming that the prior over the latent variable $Z$ is a Gaussian mixture that factorizes over unobserved discrete variable $U$, and the decoder is piecewise affine and weakly injective, then $P(U, Z)$ is identifiable from the observed data distribution $P(X)$ up to an affine transformation of $Z$.

## A.3 Implementation details

In our experiments, we trained for a certain number of epochs $M$ that depends on the size of the dataset $N$. Here are the values of $M$ and $N$ for the toy problem, as well as with VQA v2.0 and NLVR2.

Table 1: Toy problem

| $N$ | $M$ |
| --- | --- |
| 100 | 1,000 |
| 400 | 1,000 |
| 1,600 | 500 |
| 6,400 | 200 |
| 25,600 | 100 |

Table 2: VQA v2.0

| $N$ | $M$ |
| --- | --- |
| 500 | 1,000 |
| 2,000 | 1,000 |
| 8,000 | 500 |
| 32,000 | 200 |
| 128,000 | 100 |

Table 3: NLVR2

| $N$ | $M$ |
| --- | --- |
| 200 | 1,000 |
| 800 | 1,000 |
| 3,200 | 500 |
| 12,800 | 200 |
| 51,200 | 100 |

## A.4 Additional experimental results

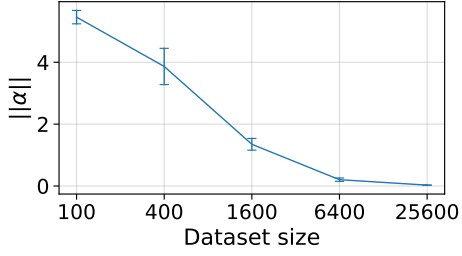
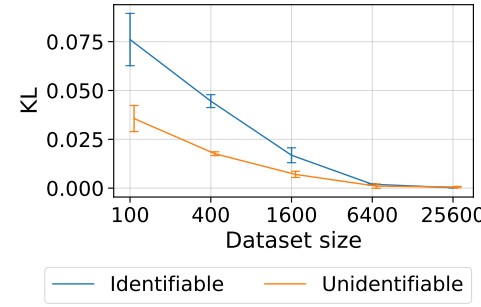

Figure 9: Detecting incidental correlation on the toy problem with $D = 16$, $b = 1.5$, and $\sigma = 0.1$.

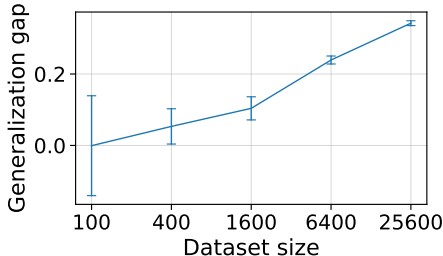

Figure 10: Comparing multimodal and unimodal learning on the toy problem with $D = 16$, $b = 1.5$, and $\sigma = 0.1$.

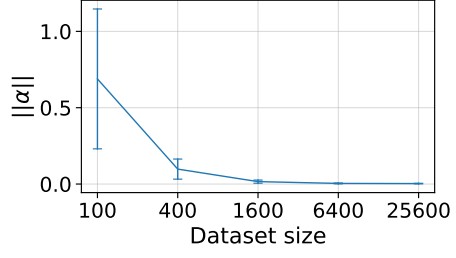
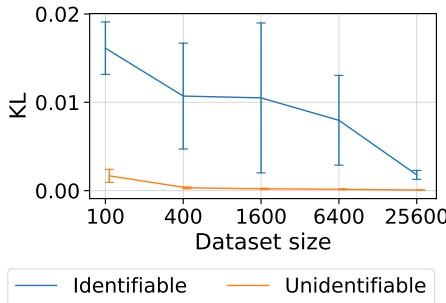

Figure 11: Detecting incidental correlation on the toy problem with $D = 1$, $b = 3$, and $\sigma = 1$.

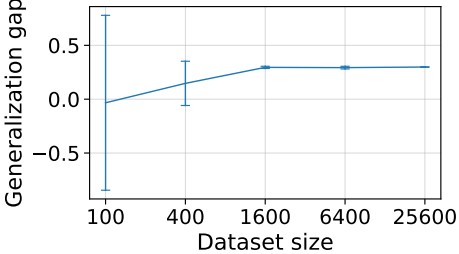

Figure 12: Comparing multimodal and unimodal learning on the toy problem with $D = 1$, $b = 3$, and $\sigma = 1$.

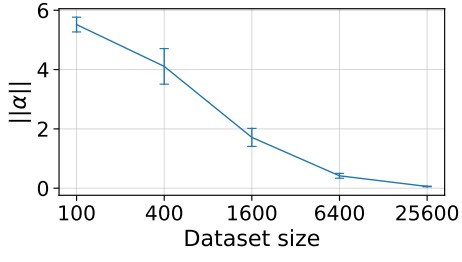
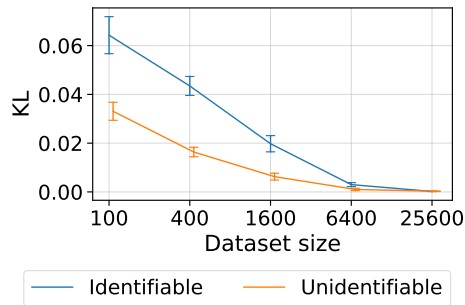

Figure 13: Detecting incidental correlation on the toy problem with $D = 16$, $b = 3$, and $\sigma = 1$.

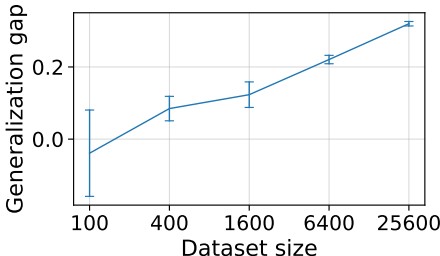

Figure 14: Comparing multimodal and unimodal learning on the toy problem with $D = 16$, $b = 3$, and $\sigma = 1$.

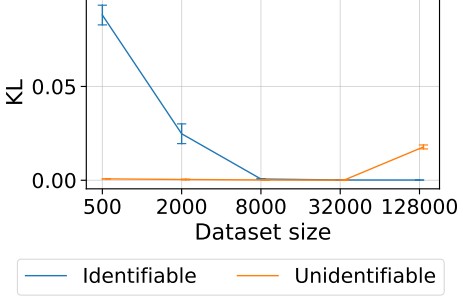
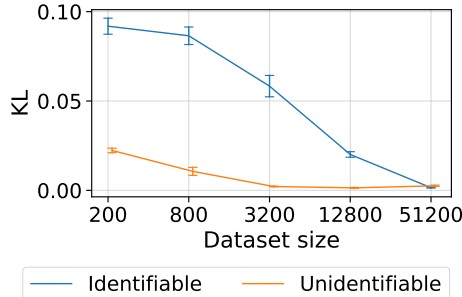

Figure 15: Detecting incidental correlation on VQA v2.0 and NLVR2 with a smaller model with latent dimension 128 and 32 components in the mixture prior.

