# OpenReview forum: "Detecting incidental correlation in multimodal learning via latent variable modeling"
_TMLR — Accepted by TMLR_

### Review · Reviewer_ARpp · 2023-05-08

**Summary Of Contributions:**

In this work, it raises a concern that not all modalities are used to boost more accurate predictions in the multimodal learning. Though the existing works address the issue by ensuring no systematic biases in the data creation or enlarging the neural network's capacity, the authors point out that such *modality underutilization* could still occur in the small data regime. They proposed an assumption that instead of the ground-truth underlying data generating process, it's more likely a different *spurious* data generating process is learnt. The authors propose an identifiable variational autoencoder, which conditions on the multimodal observations $x, x'$ and further uses target $y$ to learn the posterior for a latent variable $z$ (encoder). The decoder reads $x, x', z$ to predict the target $y$. The KL divergence between the learnt posterior and the Gaussian mixture prior is defined as the severity of incidental correlation. A toy problem with synthetic data is illustrated to demonstrate the proposed method. The identifiable VAE is also further validated on a real-world dataset VQA 2.0.

**Audience:**

Yes

**Broader Impact Concerns:**

The Broader Impact Statement section is not presented, and I don't think it is applicable to this paper.

**Claims And Evidence:**

Yes

**Requested Changes:**

1. I think the Gaussian mixture is a critical module in this work. Pls elaborate on how you define or learn the Gaussian mixture. Is it dataset-dependent?
2. It would be more convincing if you can test your method on more real-world datasets or incorporating more modalities to demonstrate its generalizability.
3. For latent variable modeling and disentangled representation learning related works, I would suggest adding [1,2] into the discussion.

[1] Li, Y. and Mandt, S., 2018, July. Disentangled sequential autoencoder. In International Conference on Machine Learning (pp. 5670-5679). PMLR.

[2] Bai, J., Wang, W. and Gomes, C.P., 2021. Contrastively disentangled sequential variational autoencoder. Advances in Neural Information Processing Systems, 34, pp.10105-10118.

**Strengths And Weaknesses:**

Pros:
1. The paper is written clearly and the motivations are clearly stated.
2. The method is concise and easy to plug in. From the description in the paper, the work could generalize to a variety of application domains.
3. One detailed toy example is provided to explain how the proposed method works and it's further validated on a real-world datasets VQA 2.0.

Cons:
1. Besides detecting whether incidental correlation is present or not, to determine the need for more data, I think another direction would be more interesting: if the dataset is large enough to get rid of the the incidental correlation issue. Furthermore, if this work can propose the potential directions of which data to further collect.
2. Small datasets are more likely to suffer from incidental correlation issue. While an unobserved var $u$ is a possibility, I would intuitively more suspect another scenario: small amount of samples are not enough to depict the data distribution, so it's not another DGP is more likely to be learnt, but instead one was "randomly" selected from multiple "eligible" DGPs. Also, I think it's a bit hard to determine if adding $u$ is enough to narrow down the DGP candidates to the ground-truth one?
3. Close-to-zero KL divergence is also often known as "over-regularization" (or posterior collapse as you mentioned). Though Gaussian mixture is proposed to alleviate the issue, how do you pre-define or learn these Gaussians? Are they learnt and defined case-by-case as in the toy example?
4. Eq. (3) has a typo in $q(z|x,x',z)$. Pls correct.
5. While I generally agree all modalities should be used if necessary, I wonder if there are some cases fewer modalities can make accurate predictions and forcing the model to use all modalities hurt the performance on the contrary?
6. You mentioned multiple modalities like text, vision, speech, etc. in the introduction. Given that you place your work as a detection method, I think it's better to test the method on more datasets.

---

> ### Author Response · Authors · 2023-05-31
>
> Thank you for taking the time to review our paper, and for your generous words regarding the clarify of the writing, the practicality and generality of our method, and the validity of our experiments. We also appreciate your suggestions for improving the paper, such as discussing active learning as a potential future direction.
>
> Below, we address your three requested changes. We will shortly release a revision that incorporates these changes.
>
> ---
>
> Requested change 1: I think the Gaussian mixture is a critical module in this work. Pls elaborate on how you define or learn the Gaussian mixture. Is it dataset-dependent?
>
> Response 1: The Gaussian mixture prior is defined in a dataset-agnostic way, and its parameters are updated while training the VAE. We specify a random variable C, which is a categorical distribution over the K mixture components. The distribution P(C) is initialized as uniform, and its parameters are updated while training. For each component, we parameterize a multivariate Gaussian with diagonal covariance matrix. The mean and the diagonal entries of the covariance are initialized as Xavier normal, and their values are also updated while training. We will include these details in the main text.
>
> [Update] See Sections 5.1.2 and 5.2.1.
>
> ---
>
> Requested change 2: It would be more convincing if you can test your method on more real-world datasets or incorporating more modalities to demonstrate its generalizability.
>
> Response 2: We agree that more results on real-world datasets can strengthen our paper, so we are running additional experiments on the NLVR2 dataset to add to the main text. This dataset represents a binary classification task to determine whether a caption is true about a pair of images.
>
> [Update] See Section 5.2.
>
> ---
>
> Requested change 3: For latent variable modeling and disentangled representation learning related works, I would suggest adding [1,2] into the discussion.
>
> Response 3: Thank you for pointing us to these papers. We agree that the discussion on disentangled representation learning can be expanded. We will add more references about unidentifiable disentangled representation learning (such as [1, 2]), note the impossibility result of Locatello et al. (2019), and follow it with recent identifiable approaches.
>
> [Update] See Section 2.3.

---

### Review · Reviewer_csjL · 2023-05-27

**Summary Of Contributions:**

Based on the observation that multimodal learning usually do not generalise better than unimodal learning, the authors hypthesizes that this is because small dataset creates the possibility of suprious correlations between an unobserved confounder in small-data regime. The authors proposed a method for detecting this \emph{incidental correlation} and showed the plausibility of their hypothesis and effectiveness of their method.

**Audience:**

Yes

**Broader Impact Concerns:**

Don't think there is any substantial concerns.

**Claims And Evidence:**

Yes

**Requested Changes:**

## Critical:
1. Please justify the use of an independent prior in the CVAE setup. Logically, u and y are conditionally independent given x and x', this implies that KL[q(z|x, x', y)||p(z|x, x')] =0, not KL[q(z|x, x', y)||p(z|)] =0. This is quite important to explain, or run experiments to justify which one is better empirically.
1. Please address this question regarding Section 4: is it the case that (at least theoretically) posterior collapse is not possible when the model is identifiable? And is it the case that the method of Kivva et al (2022) can always achieve identifiability? If so, under what conditions? Because if this is the case, then we can have conditions that the identifiable model can avoid posterior collapse under those conditions. This is conceptually an important chain of reasoning.
1.  Section 5.2.1. A Gaussian mixture with 128 components is quite large, and the dimensionality of the z-space is also huge. Can the authors run some experiments with lower dimensions? Or is it not important? This helps practitioners know what kind of dimensionalities they should expect to run to detect incidental correlations.

## Suggestions
I have a few suggestions that may help reader understand the paper better
1. typo: page 2, 2nd last para. line 4: "edge makes which makes"
2. Writing: in Section 5, it would help if the authors could allude to the readers that there will be a few experiments with different techniques (with u observed and unobserved, and comparing multipmodal vs unimodal learning). Even better, let us know what you are running these different comparisons that serve very similar purposes.
3. typo: Sec. 5.1.2, last para, first line: "show that the for the "
4. second to last line of page 8: if I understand correctly, it should be the log-probability of the "true" target, right? It doesn't hurt to clarify a bit more.
5. All figures: what are the errorbars? Are there any reasons not to run statistical tests on claims like "... is large when datasize is small"? Could we do some t-test on comparison with zero, or on the slope of the lines?


**Strengths And Weaknesses:**

## Strengths:
* The paper is super well written and I understand fully and approve of (most of) what's been done, the definition of incidental correlation and why this may hurt performance, and the broader context in which this paper may be important or limited (e.g. the case of Fig 2b).
* The experiments carefully validated their hypotheses and results are consistent across datasets and problems.
* Their method is practical and can be easily used for many problems.

## Weaknesses
* I understand the authors want to \emph{detect} incidental correlations, but do not further provide a method for improve methods that were affected by it. This is a tiny bit disappointing (although the authors are aware of it based on Conclusions).
* I feel that, although it is possible that a more complicated model (with an arrow from $u$ to $z$) can become a winning model in small-data regime, there isn't any explanation or illustration how this dependence could be. This is more out of curiosity, but helps strenghtens the authors' hypotheses.
* There is a problem with the CVAE setup: the prior does not depend on the conditioning variables.
* The distinction between the indentifiable and unidentifiable CVAEs need more explanation. The cited paper of Kivva (2022) is quite dense, so the authors should explain briefly why choosing a single, fixed prior makes the CVAE unidentifiable, or just state the assumption needed for identifiability.

---

> ### Author Response · Authors · 2023-05-31
>
> Thank you for taking the time to review our paper, and for your constructive criticism. We appreciate your kind words regarding the clarity of the writing, the validity of our experimental design, and the practicality of our method.
>
> We agree with the points that you made; here are our responses to your three requested changes that you designated as critical. We will shortly release a revision that incorporates these changes.
>
> ---
>
> Requested change 1: Please justify the use of an independent prior in the CVAE setup. Logically, u and y are conditionally independent given x and x', this implies that KL[q(z|x, x', y)||p(z|x, x')] =0, not KL[q(z|x, x', y)||p(z|)] =0. This is quite important to explain, or run experiments to justify which one is better empirically.
>
> Response 1: In our CVAE, we require z and y to be conditionally dependent given (x, x’). We agree this can be achieved with edges from z to (x, x’). However, these edges are not necessary since (x, x’) are always observed. I.e. the only open path from z to y is via the edge from z to y. Therefore, we chose remove the edges from z to (x, x’), which has the benefit of simplifying the implementation. We will clarify this in the revision.
>
> [Update] See the third to last paragraph in Section 4.
>
> ---
>
> Requested change 2: Please address this question regarding Section 4: is it the case that (at least theoretically) posterior collapse is not possible when the model is identifiable? And is it the case that the method of Kivva et al (2022) can always achieve identifiability? If so, under what conditions? Because if this is the case, then we can have conditions that the identifiable model can avoid posterior collapse under those conditions. This is conceptually an important chain of reasoning.
>
> Response 2: Yes, posterior collapse is not possible when the true model does not exhibit posterior collapse, and is identifiable up to an invertible affine transformation. The latter is achieved by Kivva et al. (2022). The reasoning is as follows. Wang et al. (2021) defined unidentifiability as p(x|z) = p(x), and posterior collapse as p(z|x) = p(z). By Bayes’ rule, the former implies the latter. Conversely, if the true model satisfies p*(x|z) ≠ p*(x), then its posterior does not collapse. If we estimate the true model up to an invertible affine transformation, then p(x|z) ≠ p(x) holds for our model, and its posterior does not collapse. In other words, a non-collapsed posterior cannot be transformed into a collapsed posterior by an invertible affine transformation. We will elaborate on this point in Section 4, and add the precise assumptions of Kivva et al. (2022) in the appendix.
>
> [Update] See the fourth paragraph in Section 4, and Section A.2.
>
> ---
>
> Requested change 3: Section 5.2.1. A Gaussian mixture with 128 components is quite large, and the dimensionality of the z-space is also huge. Can the authors run some experiments with lower dimensions? Or is it not important? This helps practitioners know what kind of dimensionalities they should expect to run to detect incidental correlations.
>
> Response 3: We will run additional experiments with less mixture components and a lower-dimensional z, and include the results in the appendix.
>
> [Update] See Fig. 15 for additional results with latent dimension 128. All of our conclusions remain intact, but the KL values are lower here, which indicates that sufficient model capacity is required to detect incidental correlation. We are currently running additional experiments with less mixture components, and will add this to the appendix when it is complete.

---

### Review · Reviewer_Ft39 · 2023-05-31

**Summary Of Contributions:**

The paper addresses a key problem in multimodal learning known as "modality underutilization," where multimodal neural networks fail to utilize all modalities, leading to weaker generalization or biased predictions. The authors introduce a concept called "incidental correlation," that leads to modality underutilization. The authors further propose a method to detect incidental correlation through latent variable modeling. They specify an identifiable variational autoencoder (VAE), allowing them to interpret the Kullback-Leibler divergence between the latent posterior and prior as the severity of incidental correlation. The paper conducts experiments with synthetic data and the VQA v2.0 dataset, validating the authors' theory of incidental correlation and demonstrating its effect on modality underutilization.

**Audience:**

Yes

**Broader Impact Concerns:**

none noted

**Claims And Evidence:**

Yes

**Requested Changes:**

Please address to the weakness listed above.

**Strengths And Weaknesses:**

- strengths
    - the paper discusses important problem of "model underutilization" and discusses a new concept called "incidental correlation".
    - the idea of using VAE to identify "incidental correlation" is interesting

- weakness
    - the writing of the paper has multiple unclear remarks, for examples
        - it is unclear how "incidental correlation" is different from the well-known terms such as spurious correlation, confounding factors, or dataset bias etc
        - the authors create multiple terms such as "Multimodal generating process" and "Spurious generating process", while it seems the authors suggest the these two terms are referring to two different scenarios mainly being differentiated by whether p(y|d(x)) equalize to p(y|x) or not, the names "Multimodal" and "Spurious" are not even orthogonal to each other.
        - It's unclear why "Small datasets that are sampled from the MGP, where u and y are conditionally independent given the inputs, can have higher likelihood under the SGP, where this independence does not hold" This seems to be a key assumption used in the paper, while this is an often observed phenomenon in practices, it does not seem to hold when the authors put into their theoretical framework, if we are talking about uniform sampling.
       - "g(u) is a function that determines the amount of information that passes from u to y given the inputs" not sure what does "amount of information" refers to.
    - the synthetic experiments setting seems to limited
       - will the synthetic results different if we test under more various settings of the parameters (1.5, 0.1)?
       - if the data are uniformally sampled, aren't Figure 4(a) and Figure 4(b) have the same error rate while using single modality? then why a model will favor one in one case over the other one in the other?
   - regarding real dataset experiment
      - there are tons of datasets that are proven to have spurious features, e.g. [1]. Probably the authors can demonstrate the usage of their methods on more datasets with known spurious features, the results might be more convincing.
      - it might be better to demonstrate accuracy-improving experiments on popular datasets.


[1]. Select-Additive Learning: Improving Generalization in Multimodal Sentiment Analysis

---

> ### Author Response · Authors · 2023-05-31
>
> Thank you for taking the time to review our paper, and for kindly saying that we have proposed an interesting idea to address the important problem of modality underutilization.
>
> Below, we address the weaknesses that you listed. We will shortly release a revision that incorporates these changes.
>
> ---
>
> Weakness 1: It is unclear how "incidental correlation" is different from the well-known terms such as spurious correlation, confounding factors, or dataset bias etc.
>
> Response 1: Incidental correlation is a specific form of spurious correlation that occurs with small datasets. See Section 1, paragraph 3 for the precise definition, as well as an intuitive example involving two independent Bernoulli random variables.
>
> ---
>
> Weakness 2: The authors create multiple terms such as "Multimodal generating process" and "Spurious generating process", while it seems the authors suggest the these two terms are referring to two different scenarios mainly being differentiated by whether p(y|d(x)) equalize to p(y|x) or not, the names "Multimodal" and "Spurious" are not even orthogonal to each other.
>
> Response 2: We chose the name “multimodal generating process” because it is the ideal setting for multimodal learning, where all modalities are properly utilized. In contrast, the “spurious generating process” is named that way because there is a spurious correlation between u and y given the inputs, which leads multimodal learning to fail.
>
> ---
>
> Weakness 3: It's unclear why "Small datasets that are sampled from the MGP, where u and y are conditionally independent given the inputs, can have higher likelihood under the SGP, where this independence does not hold" This seems to be a key assumption used in the paper, while this is an often observed phenomenon in practices, it does not seem to hold when the authors put into their theoretical framework, if we are talking about uniform sampling.
>
> Response 3: See the Bernoulli example in Section 1, paragraph 3 for intuition on how a dataset sampled from one data generating process (DGP) can have higher likelihood under an alternative DGP. In this example, the two Bernoulli variables are independent under the true DGP, but a small dataset can have higher likelihood under an alternative DGP where the variables are correlated. This statement holds true when the data is sampled uniformly, as long as the size of the dataset is small.
>
> ---
>
> Weakness 4: g(u) is a function that determines the amount of information that passes from u to y given the inputs" not sure what does "amount of information" refers to.
>
> Response 4: We will rephrase this in the revision to say that g(u) determines the degree to which u and y are conditionally independent given the inputs. On one extreme, when g(u) is constant w.r.t. u, then u and y are conditionally independent given the inputs. On the other extreme, when g(u) = y, then u and y are perfectly dependent given the inputs.
>
> [Update] See Section 3.
>
> ---
>
> Weakness 5: Will the synthetic results different if we test under more various settings of the parameters (1.5, 0.1)?
>
> Response 5: The results of the toy problem do not vary significantly w.r.t the parameters of the DGP. We will provide additional results with alternative settings in the appendix.
>
> [Update] See Fig. 9--14.
>
> ---
>
> Weakness 6: If the data are uniformally sampled, aren't Figure 4(a) and Figure 4(b) have the same error rate while using single modality? then why a model will favor one in one case over the other one in the other?
>
> Response 6: Figure 4a and 4b show two datasets sampled from the same underlying DGP, where the only difference in the number of samples. Since they come from the same DGP, you are correct that the unimodal error rate will be the same in the large data regime. However, this toy problem shows that the unimodal error rate can be artificially lower when the dataset is small. Specifically, it depends on how many red examples land in the top left quadrant, and how many blue examples land in the bottom left quadrant.
>
> ---
>
> Weakness 7: There are tons of datasets that are proven to have spurious features, e.g. [1]. Probably the authors can demonstrate the usage of their methods on more datasets with known spurious features, the results might be more convincing.
>
> Response 7: We agree that more results on real-world datasets can strengthen our paper, so we are running additional experiments on the NLVR2 dataset to add to the main text. This dataset represents a binary classification task to determine whether a caption is true about a pair of images.
>
> [Update] See Section 5.2.
>
> ---
>
> Weakness 8: It might be better to demonstrate accuracy-improving experiments on popular datasets.
>
> Response 8: Our paper describes a new failure mode for multimodal learning which we call incidental correlation, proposes a method to detect it, and detects it on real-world datasets. Improving state-of-the-art performance on benchmark datasets is an important problem, but is not within the scope of our paper.

---

### Review · Reviewer_sw4q · 2023-05-31

**Summary Of Contributions:**

This paper studies the problem of detecting spurious correlations between different data modalities in the small sample size regime. The proposed method uses a conditional VAE model with a Gaussian mixture latent space of a response conditioned on two inputs with a latent variable that represents a possible unobserved variable influencing the inputs and response. The ground truth situation is that the response is conditionally independent of the unobserved latent variable given the inputs, but it can become conditionally dependent in the small sample regime. The proposed method uses the KL term of VAE learning to as a indicator of spurious correlations from low samples sizes. High KL is interpreted to mean that spurious correlations are being represented by z, and low KL is interpreted to mean that spurious correlation are not being learned because the encoder is not needed. Experiments on synthetic data and VQAv2 show that KL decreases with sample size.

**Audience:**

Yes

**Claims And Evidence:**

No

**Requested Changes:**

* Please clarify why the vanilla VAE is presented as unidentifiable, especially with reference to Section 3.4 of Kivvas et al. 2022.
* Is there a way to apply the method to large scale datasets which might not be balanced? For example, improving the balancing on VQA v1.

**Strengths And Weaknesses:**

*Strengths:*
* The problem of understanding when multimodal models (or even unimodal models) make predictions based on spurious attributes of the data distribution is a very interesting and relevant direction.
* The intuitive direction of trying to model possible confounding information using the latent space of a VAE is a promising direction.

*Weaknesses:*
* A central part of the experiments involves comparing what is called an identifiable VAE with a learnable mixture latent space and a so-called unidentifiable standard VAE, based on the theoretical results from Kivva et al. 2022. I am not an expert on identifiable VAEs but from my understanding of the claims in Kivva et al. 2022 Section 3.4 "Classical VAE" it appears that classical VAEs are considered identifiable and a special case of the mixture VAE. Why does the paper claim that the classical VAE is unidentifiable in the context of Kivva et al.? I am confused on this point and it is central to the experiment section.
* The scope of the method and experiments seem very limited. The main method simply uses the KL divergence term of VAE learning to detect whether spurious correlations exist for small sample sizes. However, there is no heuristic presented for determining whether a sample size is sufficiently large to be free from spurious correlations. The method also does not include a constructive way of reducing spurious correlations when they are detected. Better practical applications would make the method more appealing.
* It is not entirely clear that spurious correlations are the reason why KL divergence is high for low sample sizes. Could there be other aspects of small sample sizes that cause this?
* The problem of small sample size learning in the context of multimodal models might be of limited interest. Investigating the method at scale would better display its potential.

---

> ### Author Response · Authors · 2023-06-01
>
> Thank you for taking the time to review our paper. We appreciate you saying that we are working on an interesting and relevant problem, and that our approach is intuitive and promising.
>
> We address your requested changes below, as well as the items on the list of weaknesses. We will shortly release a revision that incorporates these changes.
>
> ---
>
> Requested change 1: Please clarify why the vanilla VAE is presented as unidentifiable, especially with reference to Section 3.4 of Kivvas et al. 2022.
>
> Response 1: This was an error, and we will modify the vanilla VAE so that it violates the assumptions of Kivva et al. (2022). The two assumptions required for identifiability are that (i) the prior is a mixture, and that (ii) the decoder is piecewise affine. As you pointed out, our current vanilla VAE (unintentionally) satisfies both. Assumption (i) is satisfied because the Gaussian prior is technically a mixture with a single component, and assumption (ii) is satisfied due to our use of the ReLU activation function in the decoder. We will use the GELU activation function in the decoder in order to violate assumption (ii), and make the vanilla VAE unidentifiable.
>
> [Update] After fixing this, the unidentifiable VAE's posterior collapses frequently, and strengthens our message about the importance of identifiability. See Figures 5, 7 in the main text, and Figures 9, 11, 13, 15 in the appendix.
>
> ---
>
> Requested change 2: Is there a way to apply the method to large scale datasets which might not be balanced? For example, improving the balancing on VQA v1.
>
> Response 2: We chose VQA v2 over VQA v1 specifically because the former is designed to exhibit less systematic biases than the latter. This is important because we are showing that incidental correlation can occur in the absence of systematic bias. Since our method is a detection method, it would not be as interesting to detect dataset bias in VQA v1, since this has been extensively demonstrated by previous work.
>
> Having said that, we agree that more results on real-world datasets can strengthen our paper, so we are running additional experiments on the NLVR2 dataset to add to the main text. This dataset represents a binary classification task to determine whether a caption is true about a pair of images.
>
> [Update] See Section 5.2. All of our conclusions also hold on NLVR2.
>
> ---
>
> Weakness 1: The scope of the method and experiments seem very limited. The main method simply uses the KL divergence term of VAE learning to detect whether spurious correlations exist for small sample sizes. However, there is no heuristic presented for determining whether a sample size is sufficiently large to be free from spurious correlations.
>
> Response 1: The heuristic for determining whether a sample size is sufficiently large is to apply our method, and verify that the KL divergence between the latent posterior and prior is close to zero.
>
> ---
>
> Weakness 2: The method also does not include a constructive way of reducing spurious correlations when they are detected. Better practical applications would make the method more appealing.
>
> Response 2: The scope of our paper is to describe a new failure mode for multimodal learning which we call incidental correlation, propose a way to detect it, and detect it using real-world datasets. We agree that mitigating incidental correlation is a promising avenue for future work, and mention this in Section 6. We will expand this section in the revision to discuss concrete ways to extend our work. One promising direction suggested by reviewer ARpp is to incorporate our method in an active learning framework. I.e. we can collect additional data that maximally reduces the KL divergence between the latent posterior and prior.
>
> [Update] We added this to the last paragraph of Section 6.
>
> ---
>
> Weakness 3: It is not entirely clear that spurious correlations are the reason why KL divergence is high for low sample sizes. Could there be other aspects of small sample sizes that cause this?
>
> Response 3: The answer is no, under three conditions: (i) our causal graph (Fig 1b) is correct, (ii) the VAE is identifiable, and (iii) the VAE is trained until convergence. If our causal graph correctly describes the true DGP, then the information encoded by the latent posterior is spurious. Then, if the VAE is identifiable and is trained until convergence, the only reason why the KL term can be positive is if this spurious information is present.
>
> [Update] We added this to the last paragraph of Section 4.

---

### Decision · Action_Editors · 2023-08-29

**Recommendation:** Accept as is

**Comment:**

All reviewers thought (1) the submission is well written, and (2) the problem studied here is important. Most reviewers did not engage in the discussion, unfortunately.

Two reviewers remained unconvinced after reading the authors' responses. Two sticking points were:

(i) *The experiments could be expanded to include more real-world datasets*: a new dataset was added in the latest revision. The synthetic data were designed to study the problem in the limited data regime, so I don't think this is an issue given the scope of the claims.

(ii) *Whether the KL divergence between the posterior and prior is an appropriate metric to measure spurious correlation*: additional discussion has been added and experimental results have shown the utility of this metric.

All other requested changes have been incorporated and discussed. I recommend acceptance.

**Audience:**

Utilising multiple data sources is heavily used in practice so understanding its pitfalls, how they arise, and how to fix them are relevant.

**Claims And Evidence:**

The claims are clearly demonstrated on well-crafted synthetic experiments and two real-world datasets.